# Radiation Synthesis of Pentaethylene Hexamine Functionalized Cotton Linter for Effective Removal of Phosphate: Batch and Dynamic Flow Mode Studies

**DOI:** 10.3390/ma12203393

**Published:** 2019-10-17

**Authors:** Jifu Du, Zhen Dong, Zhiyuan Lin, Xin Yang, Long Zhao

**Affiliations:** 1School of Nuclear Technology and Chemistry & Biology, Hubei University of Science and Technology, Xianning 437100, China; duzidedu@163.com (J.D.); zhiyuanlin12318@126.com (Z.L.); sophieyangyifan@163.com (X.Y.); 2State Key Laboratory of Advanced Electromagnetic Engineering and Technology, School of Electrical and Electronic Engineering, Huazhong University of Science and Technology, Wuhan 430074, China; zhendong@hust.edu.cn

**Keywords:** radiation grafting, cotton linter, phosphate adsorption, dynamic studies

## Abstract

A quaternized cotton linter fiber (QCLF) based adsorbent for removal of phosphate was prepared by grafting glycidyl methacrylate onto cotton linter and subsequent ring-opening reaction of epoxy groups and further quaternization. The adsorption behavior of the QCLF for phosphate was evaluated in a batch and column experiment. The batch experiment demonstrated that the adsorption process followed pseudo-second-order kinetics with an R^2^ value of 0.9967, and the Langmuir model with R^2^ value of 0.9952. The theoretical maximum adsorption capacity reached 152.44 mg/g. The experimental data of the fixed-bed column were well fitted with the Thomas and Yoon–Nelson models, and the adsorption capacity of phosphate at 100 mg/L and flow rate 1 mL/min reached 141.58 mg/g. The saturated QCLF could be regenerated by eluting with 1 M HCl.

## 1. Introduction

Eutrophication means the enrichment of water in nutrients by nitrogen and phosphorus compounds, which cause an accelerated growth of algae and superior forms of vegetable life, thus leading to degradation of the aquatic ecosystem [1]. In most cases, the concentration of phosphorus is the key factor in eutrophication control. To protect eutrophication from phosphorus, regulations and guidelines of many countries have set limitations on phosphorus concentrations in discharging waters. For example, the US has recommended that the average phosphorus concentration should not surpass 0.05 mg/L in streams discharging into lakes or reservoirs [2]. Therefore, it is very important to remove low-concentration phosphorus from waters. Many physical, chemical and biological technologies were investigated for phosphate removal. Adsorption is considered an efficient technique for removing low-concentration contamination from wastewater. The advantages of adsorption are simple in design and operation, cheap to implement and effective at low concentrations [3,4].

Inorganic adsorbents (activated carbon, metal oxides, silicates, Ca and Mg carbonates), organic adsorbents (anion exchange resins) and industrial by-products (red mud, slags, fly ash) were used for phosphate removal [2]. However, these particle-based adsorbents have several disadvantages such as high cost, fragile, post-usage disposal difficulty and lower adsorption capacity [5]. The fiber-shaped adsorbents can overcome the brittleness weakness due to the radial expansion at swollen state, and have larger surface area and huge interspaces than particle adsorbents, which can improve the adsorption performance [6,7]. In recent years, the natural fibers were recognized as an efficient, cost-effective and environmentally friendly adsorbents for contamination removal purposes. Many kinds of native cellulose fiber such as wool fiber [8], cotton linter [9], protein fibers and jute fibers [10] and kapok fiber [11] have been modified by various functional groups to remove oil [12], heavy metal ions [10,13], dyes [14], Au(III) [15], fluoride and arsenic [16,17,18] and humic acid [9] from aqueous solution. Cotton linter, the relatively short fuzz left on cotton seed after the cotton ginning process, has a high cellulose content which makes it very suitable to be an adsorbent due to its biodegradability, biocompatibility and non-toxicity. However, few reports have been studied on the removal of phosphate by modified cotton linter.

Amine groups were mostly used for anion ions adsorption through electrostatic interaction [15,17]. Pentaethylene hexamine was selected as the monomer for cotton linter modification due to its abundant amino groups and excellent thermal stability. Radiation-induced graft polymerization (RIGP) has been widely used to modify various fibers with the aim to introduce various functional groups onto the cellulose substrate [19,20,21]. The adsorbents prepared by RIGP had a high adsorption velocity because the functional groups were mainly concentrated on the surface of the substrates. In this paper, a quaternary ammonium group functionalized cotton linter was prepared by RIGP and its adsorption performance to phosphate was investigated. The newly cotton linter based adsorbent prepared by RIGP are expected to have good application prospects in phosphate adsorption.

## 2. Experimental

### 2.1. Materials

Cotton linter was supplied by Jinhanjiang refined cotton Co., Ltd., (Jingmen, China). NaH_2_PO_4_, Pentaethylene hexamine (PEHA) was bought from Aladdin Chemical Co., Ltd. (Shanghai, China). HCl, NaOH, N, N-Dimethylformamide (DMF) and 1-Bromohexane were purchased from Macklin reagent Co., Ltd. (Shanghai, China).

### 2.2. Preparation of Quaternized Cotton Linter Fiber (QCLF)

The QCLF was prepared by electron beam (EB) pre-irradiation grafting technology, and the procedure is illustrated in Figure 1.

The mixture composed of 30 mL GMA, 3 mL Tween 20 and 67 mL deionized water was nitrogen flowed to get rid of oxygen. Dry cotton linter fibers (2 g) were sealed in PE bags and vacuum pumped, then the cotton linter fiber were irradiated at the dose from 10 to 50 kGy (dose rate: 10 kGy/pass) with energy of 1 MeV by an EB accelerator manufactured by Wasik Associates INC, MA, USA. After irradiation, the sample was immersed into the GMA emulsion solution for grafting reaction at 50 °C for 3 h. Then the grafted cotton linter fiber were washed and dried.

The degree of grafting (DOG) was determined by Equation (1):(1)$$\mathrm{DOG}=(\frac{W_{2}-W_{1}}{W_{1}})\times 100\%$$
where W_1_ and W_2_ were the weights of cotton linters before and after grafting, respectively.

CLF-g-GMA samples (2.0 g) with DOG 255% were immersed into 20 mL 50% PEHA with DMF solution for ring-opening reaction with the reaction condition of 80 °C and 24 h. Then, the samples were immersed into 50% 1-bromohexane with DMF solution for quaternization with the reaction condition of 70 °C and 24 h. Finally the quarterized cotton linter were washed and dried, thus QCLF was obtained.

### 2.3. Characterization

The FTIR analysis was performed in the transmittance mode on Nicolet 6700 spectrophotometer (Thermo, Waltham, MA, USA). Surface morphologies were observed by Tescan Vega3, Brno, Czech at the accelerate voltage 10 kV. Thermogravimetric Analysis (TGA) was performed by TG290F3 of Netzsch (Selb, Germany) with the temperature from room temperature to 800 °C.

### 2.4. Batch Adsorption Experiments

QCLF (0.05 g) were added into 50 mL phosphate solution shaken at 25 °C. The pH was adjusted with 0.1 mol/L HCl or 0.1 mol/L NaOH. In the adsorption kinetics study, the concentration of phosphate was 20 mg/L and the adsorption time was 2, 5, 10, 20, 30, 45, 60, 90 and 120 min. In the adsorption isotherm experiment, the concentration of phosphate was ranging from 100 to 500 mg/L with the variation intervals of 50 mg/L.

The adsorbed amount (Q_t_) of total phosphate onto QCLF was determined using Equation (2):(2)$$Q_{t}=\frac{(C_{0}{-C}_{t})\times V}{m}$$
where C_0_ and C_t_ were the phosphate concentration before and after adsorption at certain time, V was the volume of the phosphate solution and m was the mass of QCLF.

The phosphate concentration (PO_4_^−^, HPO_4_^−^ and H_2_PO_4_^−^) were determined using Ion chromatograph (MagIC Net 883, Metrohm, Switzerland).

### 2.5. Column Experiments

The column experiments were conducted with a 3 mL organic glass at a bed height of 50 mm and the inner diameter 9 mm which was fitted with QCLF (1.45 g). The inlet concentrations of phosphate passed through the column were 50 and 100 mg/L and the space velocities (SVs) were 20 to 100 h^−1^, which corresponded to flow rate 1 to 5 mL/min. After saturated adsorption, 1 mol/L HCl was used to regenerate the QCLF at flow rate 0.25 mL/min.

## 3. Results and Discussion

### 3.1. Synthesis of the QCLF

GMA was grafted onto cotton linter by radiation grafting technology. Figure 2 shows the effect of radiation dose on the DOG in 30 wt% GMA solution. The DOG was firstly increased with dose up to maximum value of 255% at 30 kGy, and then maintained at a high platform. This phenomena can be explained by the decay mechanism of the trapped radicals. The grafting of GMA onto cotton linter by EB pe-irradiation graft polymerization was mainly initiated by the free radical mechanism. A high adsorbed dose can initiate more amounts of free radicals and induce high DOG. However, a higher absorbed dose may also result in the decomposition of cellulose substrate. So in this study, the grafted fibers with the DOG of 255% at 30 kGy were used for further experiments.

The grafted cotton linter fiber (CLF-g-GMA) with DOG 255% was immersed into PEHA solution for epoxy ring-opening reaction. The conversion rate calculated by the mass increase was 45%. The quaternization rate in the quaternization reaction was 48% calculated by mass increase.

### 3.2. Characterization

#### 3.2.1. FT-IR Analysis

Figure 3 shows the FT-IR spectra of the original cotton linter (a), CLF-g-GMA (b), CLF-g-GMA-PEHA (c), and QCLF (d). The bands of cellulose observed in curve (a) included O–H, C–H, H–O–H, C–O, C–O–C bonding at 3345, 2920, 1635, 1060 and 898 cm^−1^, respectively [22]. The grafting of GMA was confirmed by the adsorption bands at 1726 cm^–1^ and 908 cm^−1^ which was assigned to the stretching of the carbonyl and epoxy group of GMA. After the ring-opening reaction, the bands at 1564 cm^−1^ and 1465 cm^−1^ can be assigned to N–H and C–H bands of PEHA [23]. After the quaternization reaction, the peaks at 1656 cm^−1^ and 3415 cm^−1^ attributed to the appearance of quaternary nitrogen and then the intensity of N–H peaks sharply decreased, meaning that tertiary amines were converted to quaternary ammonium [24,25]. The bands at 2960 cm^−1^ were assigned to C–H antisymmetric and symmetric stretching of –CH_2_–, indicating successful introduction of alkyl chain from 1-bromohexane.

#### 3.2.2. SEM Photographs

The surface morphologies of cotton linter (a), CLF-g-GMA (b), CLF-g-GMA-PEHA (c) and QCLF (d) are shown in Figure 4. The diameters of these fiber sample increased after grafting and further modification, which might be due to the large DOG of GMA on the cotton linter.

#### 3.2.3. TG Analysis

Figure 5 shows the thermal stability of cotton linter, CLF-g-GMA, CLF-g-GMA-PEHA and QCLF. The weight loss below 100 °C was due to the loss of water in the samples. The weight loss of the origin cotton linter happened in the range from 340 to 420 °C, which showed a one-step weight loss. The weight loss of CLF-g-GMA occurred at 220 °C and terminated at 420 °C, which was due to the complex thermal decomposition of the cotton linter and grafted epoxy groups and ester in GMA. The decrease of the degradation temperature for cellulose after grafting of GMA also happened in reference [26]. But for CLF-g-GMA-PEHA and QCLF samples, the thermal degradation curves have two districts. The main decomposition temperature occurred between 220 and 450 °C, the second loss weight took place at 450 °C, which might be due to the residual mass of solid carbon. It suggested that the adsorbents have good thermal resistance for application in phosphate removal.

### 3.3. Phosphate Adsorption in Batch Experiments

#### 3.3.1. pH Study

The adsorption of phosphate onto QCLF at different initial pH values are shown in Figure 6. The adsorption capacity was little affected at pH ranging from 4 to 8. The adsorption capacity of QCLF for phosphate was very low at pH < 4 because the dominant phosphate species at lower pH were transformed to neutron H_3_PO_4_, which had lower affinity to the adsorption sites of the QCLF [27,28]. At pH between 4 and 8, the dominant species was transformed into H_2_PO_4_^−^ and HPO_4_^2−^. The adsorption capacity of QCLF for phosphate was dropped significantly at pH > 8 because there would be a competition adsorption between OH^−^ and phosphate. It was reported that the pH value in the eutrophic lake was in the range 7.5 to 8.5 [2]. So QCLF was very suitable for phosphate removal in eutrophic lakes. The phosphate concentration at pH 7 without pH adjustment was used for further batch and column adsorption tests.

#### 3.3.2. Effect of Adsorbent Dosage

Adsorption dosage was a vital parameter influencing the adsorption performance. Figure 7 shows the removal efficiency and adsorption capacity of phosphate at different adsorbent dosages. It was evident that Q_e_ decreases while the removal % increases significantly when the amount of QCLF increases. This can be explained by the fact that more adsorption sites of adsorbent were worked for electrostatic attractive forces between QCLF and phosphate. The removal efficiency reached 90% at 1 and 92% at 1.4 g/L, which was a smaller change compared to when the dosage was greater than 1 g/L. Therefore, the effectiveness decreased with an increase of adsorbent beyond 1 g/L. So the adsorbent dosage of 1 g/L (0.05 g QCLF in 50 mL solution) was regarded as the optimal dosage for further batch experiment condition.

#### 3.3.3. Adsorption Kinetics

The adsorption equilibrium time is of great importance for studying the affinity of the adsorbents to phosphate. The experimental data was fitted by three kinetics models. The pseudo-first-order, pseudo-second-order and intra-particle diffusion kinetic equations were expressed by Equations (3)–(5), respectively [29].
(3)$$\ln(q_{e}-q_{t})=\ln q_{e}-k_{1}t$$
(4)$$\frac{t}{q_{t}}=\frac{1}{k_{2}q_{e}^{2}}+\frac{t}{q_{e}}$$
(5)$$q_{t}=k_{\mathrm{id}}t^{1/2}+I$$
where q_t_ (mg/g) was the adsorption capacity of phosphate at time t (min) and q_e_ was the adsorption capacity at equilibrium time. k_1_ and k_2_ were the rate constant. K_id_ parameter was the reaction rate constant (mg/g·min^1/2^) and *I* was the intercept.

The adsorption capacity at different adsorption times is shown in Figure 8a. It can be seen that the adsorption capacity increased rapidly and reached equilibrium within 120 min. The pseudo-first-order and pseudo-second-order were used to analyze the experimental data. The parameters q_e_, k_1_ and k_2_ were calculated and are listed in Table 1. Figure 8b shows the plots of pseudo-second-order kinetic model. The higher correlation coefficient R^2^ (0.9998) indicated that pseudo-second-order kinetic model can describe the adsorption kinetics well, which indicated the adsorption of phosphate onto QCLF was a monolayer adsorption and chemical adsorption.

Intra-particle diffusion model was also used to study the transportation of phosphate to QCLF. Figure 8c shows the plot of intra-particle diffusion model. The plot were divided into two distinct regions. The first region was a fast adsorption and the second region corresponded to a slow equilibrium adsorption. And the fitted plot did not pass through the origin, indicating that intra-particle diffusion was not the only rate-controlling step [29]. The parameters of the intra-particle kinetic model are listed in Table 1.

#### 3.3.4. Adsorption Isotherms

The adsorption isotherm of QCLF was conducted at 25 °C with the phosphate concentration of 100–500 mg/L. Figure 9a shows the equilibrium adsorption capacity at different equilibrium concentrations after adsorption. The equilibrium adsorption capacity increased and finally attained the maximum value.

The Langmuir model was applicable to adsorption on homogeneous surface, which was expressed by Equation (6). Equilibrium parameter (R_L_) reflected the nature of the dominant adsorption mechanisms in the studied system. The favorability of the adsorption was given by the dimensionless separation faction R_L_, which were calculated from Equation (7):(6)$$\frac{\mathrm{Ce}}{q_{e}}=\frac{Ce}{q_{m}}+\frac{1}{K_{L}q_{m}}$$
(7)$$R_{L}=1/(1+K_{L}C_{0})$$
where C_e_ (mg/L) was the phosphate concentration after adsorption, q_e_ and q_m_ (mg/g) were the equilibrium and theoretical maximum adsorption capacity. K_L_ (L/mg) was the Langmuir constant related to the adsorption affinity [30].

The Freundlich model assumed a heterogeneous surface and its linear form was expressed by Equation (8).
(8)$$\ln q_{e}=\ln K_{F}+\frac{1}{n}\ln C_{e}$$
where, K_F_ was the Freundlich constant related to adsorption capacity (mmol/g) and 1/n represent the adsorption intensity of adsorbate on adsorbent [30].

The linear equation for Temkin model was given as Equation (9) [31].
(9)$$q_{e}=B_{t}\ln(K_{t})+B_{t}\ln(C_{e})$$
where B_t_ was related to the heat of sorption and K_t_ (L/mg) was the equilibrium binding constant.

The linear fitted plot of Langmuir, Freundlich isotherm and Temkin model are shown in Figure 9b–d, whose key parameters are summarized in Table 2. Comparatively, the Langmuir isotherm model with high R^2^ value (0.9952) was a better fit for the adsorption of phosphate on QCLF, which denoted the monolayer adsorption of phosphate onto QCLF. The adsorption capacity Q_m_ were calculated to be 152.44 mg/g, which was higher than other literature reports regarding adsorbents, especially at pH 6–7 (see Table 3). The calculated R_L_ in this study were between 0 and 1, which reflected the favorable adsorption of phosphate [31,32].

### 3.4. Column Experiments

Adsorption was an accumulation process of the adsorbate species on the absorption sites of the adsorbent. In the fixed-bed column test, the solution flows continuously through a column of adsorbent. The adsorbent near the column inlet will be saturated first, then the adsorption zone is moved further toward the exit of the bed. When all the adsorbents are saturated, the value of leakage concentration is increased and finally reaches the influent concentration. A plot of leakage concentration of the outlet as a function of proceed time was known as the breakthrough curve. In this study, column tests were carried out in a fixed-bed column by varying different initial concentrations and flow rate. The aim of this experiment was to find the optimal flow rate and influent phosphate concentration to maximize the productivity and efficiency.

#### 3.4.1. Effect of Influent Concentration

The effect of different phosphate concentration on breakthrough performance is shown in Figure 10a,b. The mass of the adsorbent and flow rate were kept constant. The efficiency of phosphate removal was described by the percent of (C_t_/C_0_). C_t_/C_0_ in all cases was increased with the time increased, until it reached 1.0. The breakthrough point and the saturated time of phosphate adsorption depended on various influent phosphate concentration. At higher influent concentration, an earlier breakthrough point was observed. Breakthrough point was decreased from 2610 to 1260 min with the influent concentration increased from 50 to 100 mg/L. At higher influent concentration, the function groups were rapidly combined with phosphate which resulted in a decrease in breakthrough time. The driving force increased with the influent concentration increased, then the adsorption points were saturated faster with higher influent concentration.

#### 3.4.2. Effect of Flow Rate

The effect of flow rate at the same influent concentration is compared in Figure 10b,c. It can be seen that the breakthrough points happened at 1260 and 200 min at flow rate of 1 and 5 mL/min, respectively. Their saturation adsorption was reached at 3200 and 500 min at flow rate of 1 and 5 mL/min, respectively, which showed that a high flow rate will saturate the fixed column faster. The difference in the slope of the breakthrough curve and adsorption capacity between the two plots may be explained on the basis of mass transfer fundamentals. At a higher flow rate, phosphate has a short diffusion time and interacts predominately with the functional groups on QCLF which have the highest availability. In regard to a lower flow rate of influent, phosphate had more time to contact with larger numbers of function groups, and this resulted in the higher removal of phosphate in the column.

#### 3.4.3. Estimation of Breakthrough Curve

Bohart–Adams, Thomas and Yoon–Nelson models were used for describing and analyzing the laboratory-scale column, which linear forms were presented in the following Equations (10)–(12), respectively [4,38,39].
(10)$$\ln(\frac{C_{t}}{C_{0}})=k_{AB}C_{0}t-k_{AB}N_{0}\frac{Z}{F}$$
(11)$$\ln(\frac{C_{t}}{C_{0}}-1)=\frac{K_{TH}q_{0}m}{v}-K_{TH}C_{0}t$$
(12)$$\ln(\frac{C_{t}}{C_{0}{-C}_{t}})=k_{YN}t-\tau k_{YN}$$
where C_0_ was the influent and C_t_ was the leakage concentration (mg/L) at time t; k_AB_, k_TH_ and k_YN_ were the model constants; t was the processing time; q_0_ was the adsorption capacity; v was the flow rate; m was the adsorbent mass; N_0_ is the saturation concentration; Z was the bed depth of column and F was the superficial velocity defined as the ratio of the volumetric flow rate Q to the cross-sectional area of the bed A. τ was the time for 50% adsorbent saturated. The key parameters of the three models are fitted, calculated and given in Table 4. From the value of the linear fitting correlation coefficient R^2^ in Table 4, both the Thomas and Yoon–Nelson models can be shown to have better predicted the adsorption performance for adsorption of phosphate in a fixed-bed column [38].

The Bohart–Adams model was established by assuming that the adsorption equilibrium is not instantaneous and that the rate of adsorption is proportional to both the residual capacity of the adsorbent and the concentration of the adsorbate species. It can be seen from Table 4 that the kinetic constant k_AB_ decreased as the phosphate concentration increased but increased as the flow rate increased. The saturation concentration (N_0_) of the column also increased as the phosphate concentration increased but decreased as the flow rate increased. Because higher N_0_ and lower k_AB_ means less adsorption resistance, the optimal performance of column experiment were then obtained at higher initial phosphate concentrations and lower flow rate [39].

According to the Thomas model, at the same flow rate of 1 mL/min, the adsorption capacity of phosphate reached 120.61 and 141.58 mg/g at 50 and 100 mg/L, respectively. The results demonstrate that the maximum adsorption capacity was increased with the increasing of the phosphate concentration. This was mainly because the adsorption force increased when the concentration of phosphate increased. On the other hand, the value of Thomas constant K_TH_ was high at lower influent concentration. The adsorption capacity also decreased significantly when the flow rate increased from 1 to 5 mL/min. Furthermore, the adsorption capacity (q_0_ = 141.58 mg/g) was slightly smaller than that of theoretical maximum capacity in batch experiment (Q_m_ = 152.44 mg/g), which showed the column test condition (100 mg/L, 1 mL/min) was suitable to practical use. According to the Yoon–Nelson model, the time τ for penetration to reach 50% decreased significantly with the influent concentration from 50 to 100 mg/L at the same flow rate or with the flow rate from 1 to 5 mL/min at the same concentration of 100 mg/L. All three τ values were very close to the experimental results. So it can be concluded that a higher maximum adsorption capacity can be obtained with a high concentration and low flow rate from both the Thomas and Yoon–Nelson model.

#### 3.4.4. Elution Experiment

After the saturated adsorption at concentration 100 mg/L and flow rate 1 mL/min, the phosphate loaded QCLF was regenerated to realize repeated use. Figure 11 shows the elution curves with 1 M HCl at flow rate 0.25 mL/min, the elution process was almost completed within 60 min, which showed that QCLF can be effectively regenerated and for further reuse.

## 4. Conclusions

A QCLF adsorbent was successfully synthesized by radiation induced grafting technique. The adsorption kinetics of QCLF for phosphate reached equilibrium within 120 min and the kinetics was well obeyed pseudo-second-order mode. The adsorption isotherms were well obeyed Langmuir model with the maximum adsorption capacity 152.44 mg/g. Column experiments showed that the breakthrough curves were dependent on initial phosphate concentration and flow rate. The saturated adsorbents could be efficiently generated by eluting with 1 mol/L HCl. The Thomas and Yoon–Nelson models were both successfully used to predict the breakthrough curves. The adsorption capacity of phosphate at 100 mg/L and flow rate 1 mL/min reached 141.58 mg/g according to the Thomas model. The results showed that the modified cotton linter fibers can be used for phosphate removal. On the other hand, the performance of the fibers after repeated use or magnified use need further investigation in the future.

## Figures and Tables

**Figure 1 materials-12-03393-f001:**
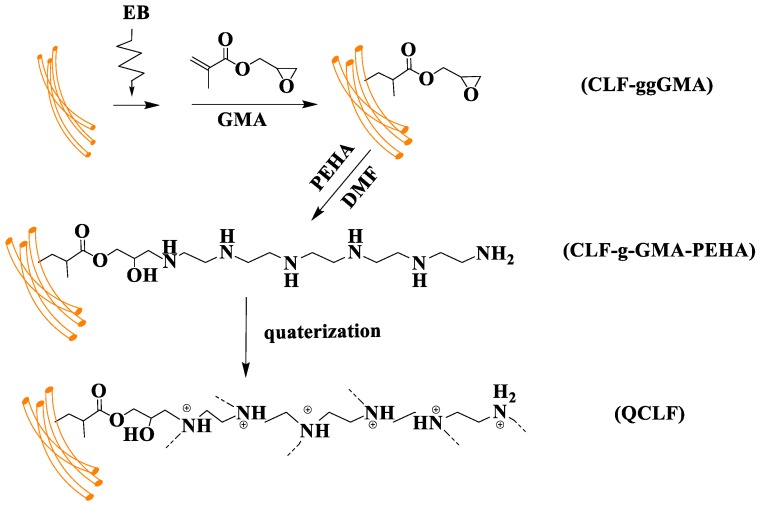
Synthesis route of quaternized cotton linter fiber (QCLF) adsorbent.

**Figure 2 materials-12-03393-f002:**
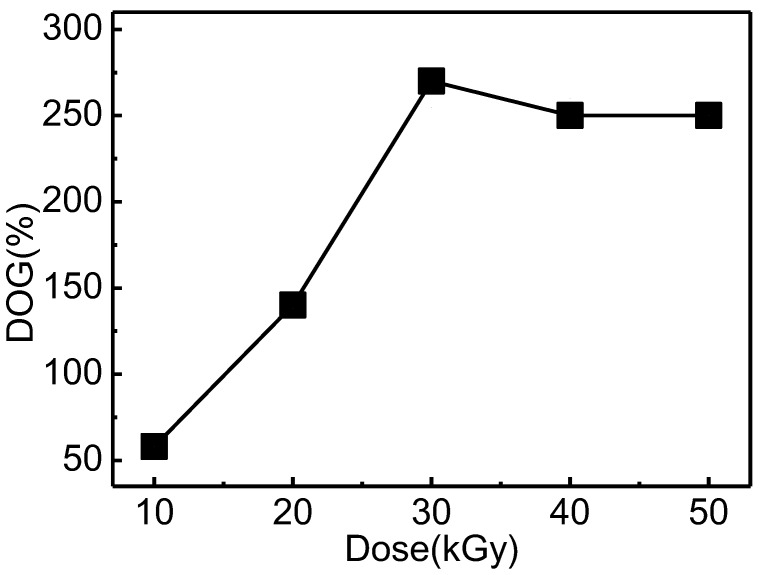
Effect of radiation dose on the degree of grafting (DOG).

**Figure 3 materials-12-03393-f003:**
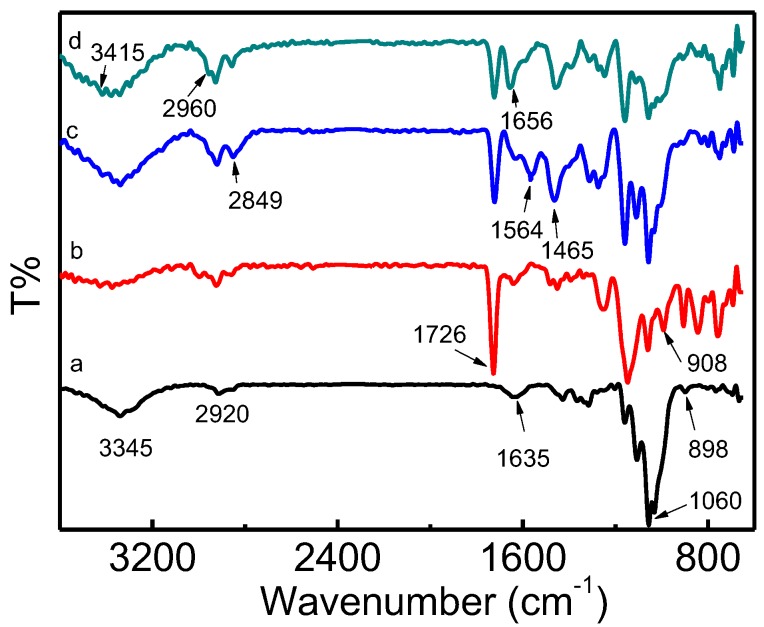
FT-IR spectra of original cotton linter (a), grafted cotton linter fiber (CLF-g-GMA) (b), CLF-g-GMA-Pentaethylene hexamine (PEHA) (c) and QCLF (d).

**Figure 4 materials-12-03393-f004:**
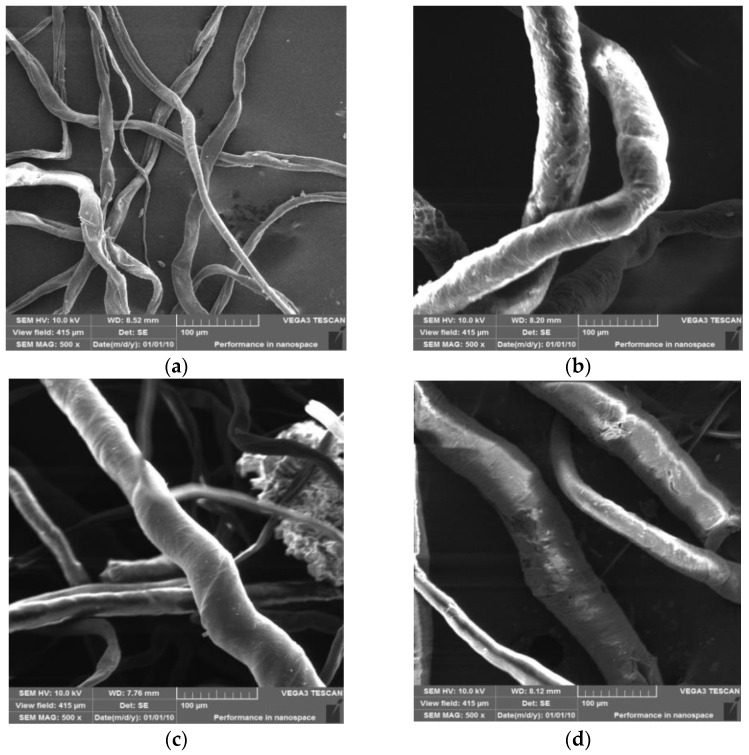
SEM photographs of original cotton linter (**a**), CLF-g-GMA (**b**), CLF-g-GMA-PEHA (**c**) and QCLF (**d**).

**Figure 5 materials-12-03393-f005:**
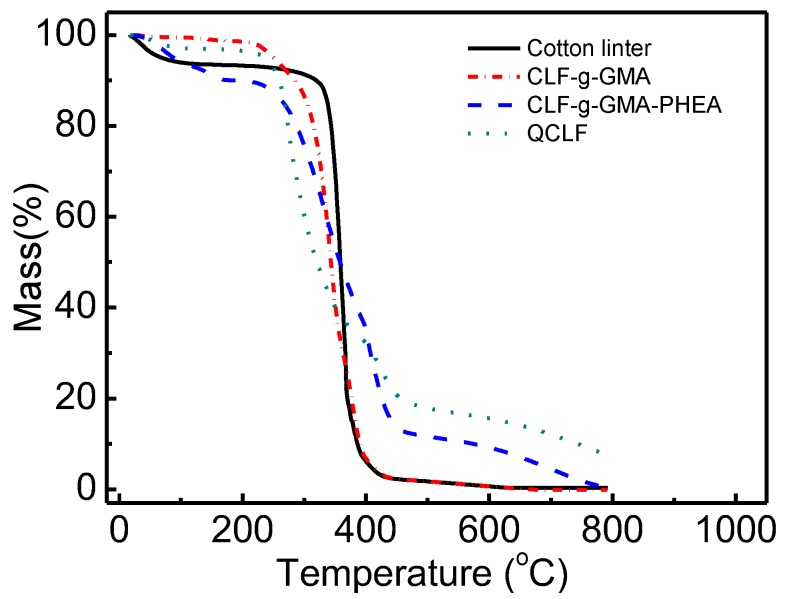
TG analysis for cotton-linter, CLF-g-GMA, CLF-g-GMA-PEHA and QCLF.

**Figure 6 materials-12-03393-f006:**
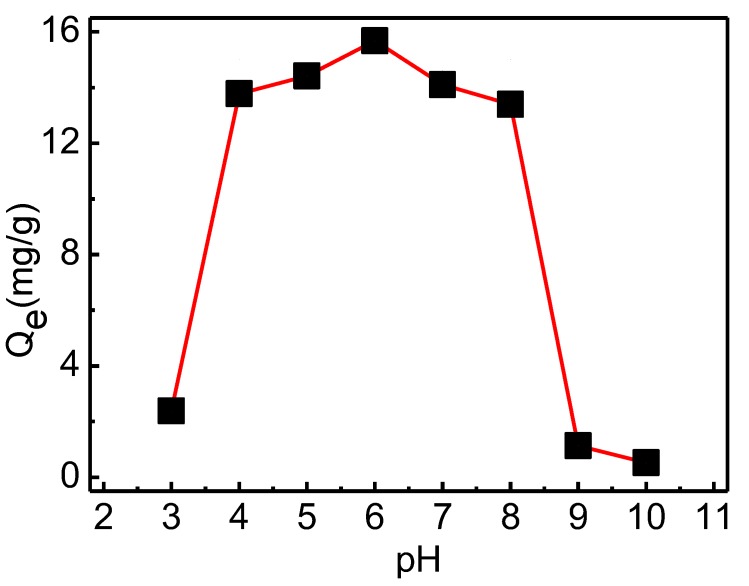
Effect of pH on phosphate adsorption by QCLF adsorbents (initial phosphate concentration 20 mg/L, volume 50 mL, mass 0.05 g, adsorption time 24 h).

**Figure 7 materials-12-03393-f007:**
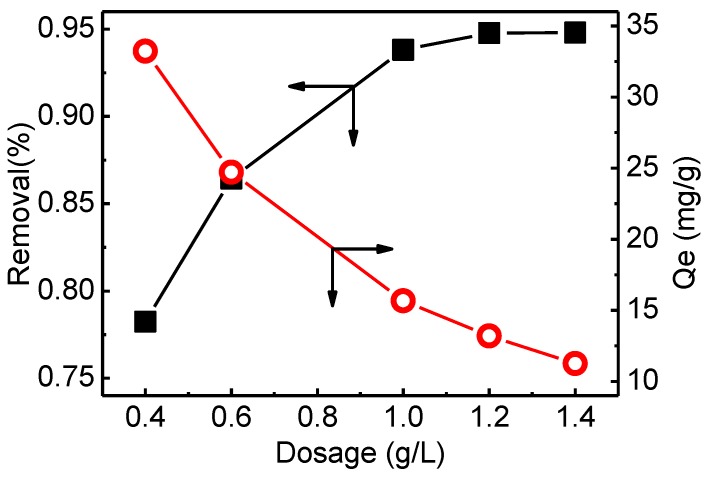
Effect of adsorbent dosage on the removal efficiency and adsorption capacity of phosphate by QCLF adsorbent (initial concentration 20 mg/L, volume 50 mL, pH 7, adsorption time 24 h).

**Figure 8 materials-12-03393-f008:**
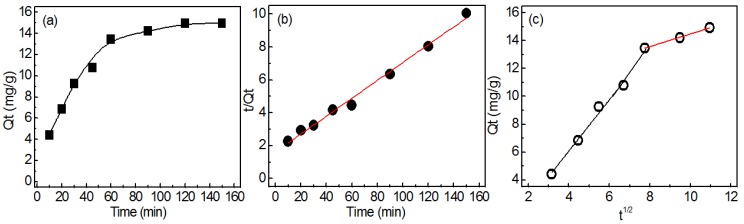
Adsorption kinetics of phosphate removal onto QCLF in 50 mL, 20 mg/L, 0.05 g, pH: 7, (**a**) effect of adsorption time; (**b**) pseudo-second-order kinetic model; (**c**) intra-particle diffusion model.

**Figure 9 materials-12-03393-f009:**
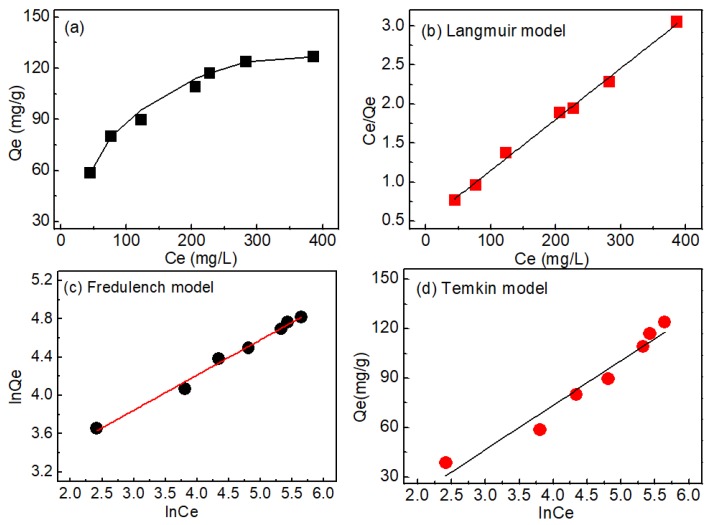
Equilibrium studies of phosphate adsorption: relation of the adsorption capacity with the equilibrium concentration (**a**); Langmuir isotherm model (**b**); Freundlich isotherm model (**c**) and Tekmin isotherm model (**d**).

**Figure 10 materials-12-03393-f010:**
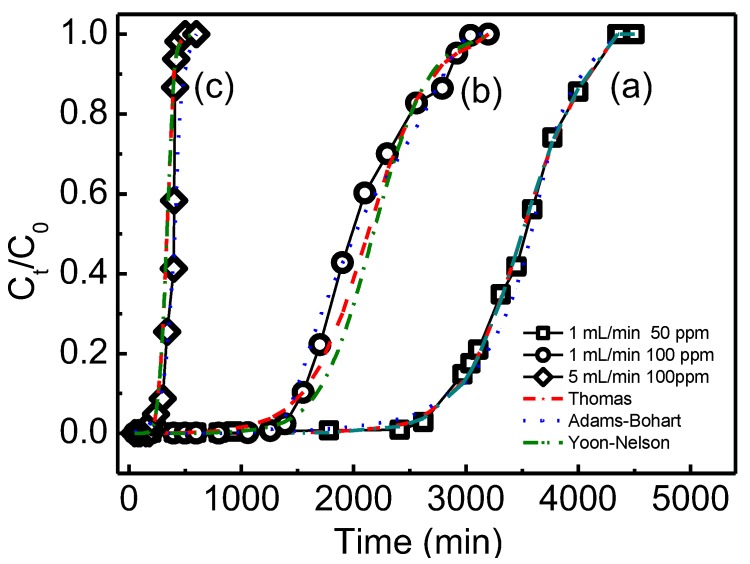
Breakthrough curves with concentration 50 mg/L at flow rate 1 mL/min (a), with concentration 100 mg/L at flow rate 1 mL/min (b), and with concentration 100 mg/L at flow rate 5 mL/min (c).

**Figure 11 materials-12-03393-f011:**
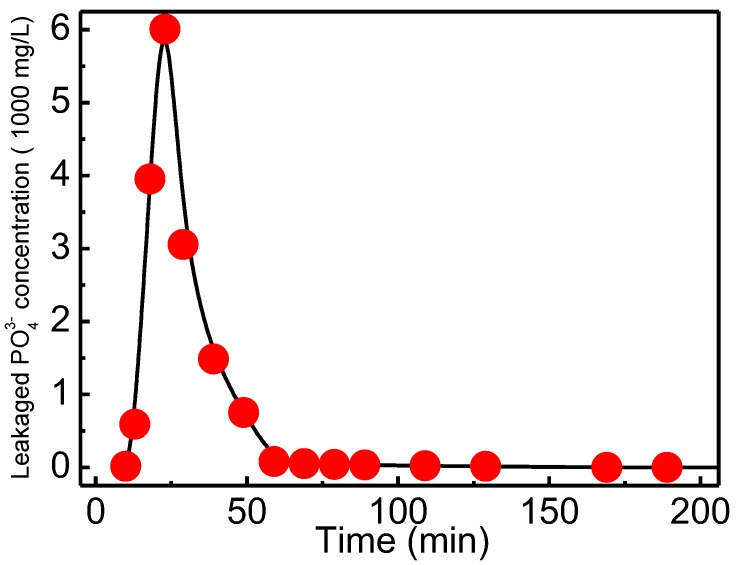
The elution curve at flow rate 0.25 mL/min.

**Table 1 materials-12-03393-t001:** Kinetic parameters obtained from pseudo-first-order, pseudo-second-order kinetic and intra-particle diffusion model.

Model	Parameters	20 mg/L HPO_4_^2−^
pseudo-first-order kinetics	k_1_ (h^−1^)	0.0305
	q_e_ (mg/g)	15.314
	R^2^	0.9872
pseudo-second-order kinetics	k_2_ (g/(mg·min))	0.0047
	q_e_ (mg/g)	19.409
	R^2^	0.9967
Weber–Morris	Ki_d1_	1.8415
	I_1_	1.3091
	R^2^	0.97891
	Ki_d2_	0.4543
	I_2_	9.9282
	R^2^	0.9973

**Table 2 materials-12-03393-t002:** Langmuir, Freundlich, Tekmin isotherm model parameters and correlation coefficients for the adsorption of phosphate.

Adsorbent	Langmuir			Freundlich			Temkin		
	Q_m _(mg/g)	K_L_	R^2^	K_F _(mg·L^−1^)	n	R^2^	B_T_	K_T_	R^2^
QCL	152.44	0.0139	0.9952	15.52	2.720	0.9899	26.9827	0.2799	0.9476

**Table 3 materials-12-03393-t003:** Comparison of adsorption capacity of QCLF with other available different adsorbents.

Adsorbent	The max Adsorption Capacity (mg/g)	pH	Reference
carbonized sludge adsorbent	4.792	7	[32]
diethylamine modified Cellulose	22.88	6.8	[33]
humic acid coated magnetite nanoparticles	28.9	6.6	[34]
quaternized pectin	31.07	7	[30]
wheat straw anion exchanger	52.80	-	[35]
amine-crosslinked Shaddock Peel	59.89	3	[27]
Zirconium (IV) loaded cross-linked chitosan particles	71.68	3	[36]
modified sugarcane bagasse fibers-Fe	152	3	[37]
quaternized cotton linter fiber	152.44	7	This paper

**Table 4 materials-12-03393-t004:** Parameters obtained from three dynamic adsorption models.

C_0_ (mg/L)	v (mL/min)	Thomas Model			Bohart-Adams Model			Yoon-Nelson Model		
		k_TH_	q_0_	R^2^	k_BA_	N_0_	R^2^	k_YN_	τ	R^2^
50	1	0.0738	120.61	0.9889	0.0462	61.1088	0.8930	0.0037	3497.81	0.9889
100	1	0.0376	141.58	0.9352	0.0102	89.229545	0.8009	0.0038	2123.74	0.9352
100	5	0.3458	22.419	0.9399	0.1363	14.4303	0.9399	0.0346	336.63	0.9368

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
