# Peer review of "Radiation Synthesis of Pentaethylene Hexamine Functionalized Cotton Linter for Effective Removal of Phosphate: Batch and Dynamic Flow Mode Studies"

_materials, 2019, doi:10.3390/ma12203393_

Round 1

Reviewer 1 Report

The present study attempts to provide an effective method for removal of phosphate from water sources using quaternized cotton linter fibers. Comparing this adsorbent with the previously studied methods, as shown in Table 3, promises a significant improvement in the maximum adsorption capacity.

Overall, the manuscript is reasonably laid out, however, in some sections, the presentation of the data lacks scientific sophistication and should be improved.

Please find below a list of my detailed comments:

More detailed discussion on the use of natural fibers for contamination removal purposes should be provided in the introduction. scientifically speaking, there is no point in fitting a curve to the data (as done in Figure 2, Figure 6, etc.) if its representing equation is not mentioned. If the whole purpose of the curve is to show a trend, it could easily be achieved by connecting data points with lines. Figure 4: the letter d in parentheses should be added at the end of the figure caption. It would be beneficial to the readers to include references for all the equations mentioned in the manuscript. Some recommendations for future work could be added. Before consideration for publication, I would recommend the authors to use a professional language editing service.

In summary, I think that the paper fits the scope of the journal and after applying these changes, it would be ready for publication.

Author Response

Dear editor

We have revised our manuscript carefully according to the reviewer’s suggestion. Our responses to the comments of the referees are described in the following paragraphs.

1# More detailed discussion on the use of natural fibers for contamination removal purposes should be provided in the introduction.

Answer: Thank you for your kind suggestion, we have rewritten the introduction section to introduce natural fibers for contamination removal.

2# scientifically speaking, there is no point in fitting a curve to the data (as done in Figure 2, Figure 6, etc.) if its representing equation is not mentioned. If the whole purpose of the curve is to show a trend, it could easily be achieved by connecting data points with lines.

Answer: Thank you for your kind suggestion, we have revised the two figures.

3# Figure 4: the letter d in parentheses should be added at the end of the figure caption.

Answer: Thank you for your kind suggestion, we have added the letter d in the caption of Figure 4.

4# It would be beneficial to the readers to include references for all the equations mentioned in the manuscript.

Answer: Thank you for your kind suggestion, we have checked the references for all the equations and added the reference.

5# Some recommendations for future work could be added.

Answer: Thank you for your kind suggestion, we have added it at the end of the manuscript. The results showed that the modified cotton linter fibers can be used for phosphate removal. On the other hand, the performance of the fibers after repeated use or magnified use need further investigation in the future.

6#  Before consideration for publication, I would recommend the authors to use a professional language editing service.

Answer: Thank you for your kind suggestion. We tried our best to improve the manuscript and made some changes in the manuscript follow the comments and the similarity report. These changes will not influence the content and framework of the paper. And here we did not list the changes but marked in red in revised paper.

Once again, Thanks very much for your comments and suggestions. We are looking forward to your positive response.

Thanks and best regards.

Sincerely yours,

Long Zhao

Reviewer 2 Report

Manuscript entitled “Radiation synthesis of pentaethylene hexamine functionalized cotton linter for effective removal of phosphate: batch and dynamic flow mode studies submitted by Jifu Du, Zhen Dong, Zhiyuan Lin, Xin Yang and Long Zhao, can be accepted for publishing in the Materials Journal, after a major revision.

In this study, a quaternized cotton linter fiber was used as adsorbent for removal of phosphate from aqueous media. The adsorbent was prepared by grafting glycidyl methacrylate onto cotton linter and subsequent ring-opening reaction of epoxy groups and further characterized. The manuscript presents original results, which are relatively well organized and systematic, but their interpretation is very poor, and this aspect should be significantly improved before the manuscript is accepted for publication.

            Here is a list of my specific comments:

General comment: All the experimental results should be more detailed interpreted and explained. Line 27: “…when phosphorus in water exceeds 0.01 mg/L.” Add here as reference the paper: Evolution of trophic parameters from Amara Lake, Environmental Engineering and Management Journal, 14(3), (2015), 559-565, because it is relevant for this observation. 2.4. Batch adsorption experiments: The variation intervals of each experimental parameter analyzed in this study should be mentioned here. Lines 122-123: ‘In curve (d), the adsorption peak of N-H was disappeared and 1656 cm-1 appeared due to quaterization reaction.” This observation should be more detailed explained. 3.2.3. TG Analysis: The utility of these experimental results should be highlighted more. Line 157: “The origin phosphate concentration at pH 7…” How was obtained this value of pH??? 3.3.3. Adsorption kinetics: The experimental results included in this section should be more detailed interpreted. 3.3.4. Adsorption isotherms: The same observation as above. 3.4. Column experiments: Again, the experimental results included in this section are too brief presented. Some additional comments/observation should be included to obtain a properly interpretation of these results. Please see the paper: Sorption of Pb(II) onto a mixture of algae waste biomass and anion exchanger resin in a packed-bed column, Bioresource Technology, 129, (2013), 374–380, which can be useful in the results interpretation.

Author Response

Dear editor

We have revised our manuscript carefully according to the reviewer’s suggestion. Our responses to the comments of the referees are described in the following paragraphs.

1# General comment: All the experimental results should be more detailed interpreted and explained.

Answer: Thank you for your kind suggestion, we have revised our manuscript and explained the experimental results more detail.

2#  Line 27: “…when phosphorus in water exceeds 0.01 mg/L.” Add here as reference the paper: Evolution of trophic parameters from Amara Lake, Environmental Engineering and Management Journal, 14(3), (2015), 559-565, because it is relevant for this observation.

Answer: Thank you for your kind suggestion, reference [1] have been added.

3# 2.4. Batch adsorption experiments: The variation intervals of each experimental parameter analyzed in this study should be mentioned here.

Answer: Thank you for your kind suggestion. We have described the experimental parameter in detail.

 In the adsorption kinetics study, the concentration of phosphate was 20 mg/L and tested time was 2, 5, 10, 20, 30, 45, 60, 90 and 120 min. In the adsorption isotherm experiment, the concentration of phosphate was ranging from 100 to 500 mg/L with the variation intervals of 50 mg/L. We have revised it in the experimental section. Also the dose rate of radiation grafting was added (dose rate: 10 kGy/pass) (Section 2.2).

4# Lines 122-123: “In curve (d), the adsorption peak of N-H was disappeared and 1656 cm-1 appeared due to quaterization reaction.” This observation should be more detailed explained.

Answer: Thank you for your kind suggestion. We have referenced more published papers and explained the IR spectra in detail.

After quaternization reaction, the peaks at 1656 cm-1 and 3409 cm-1 were attributed to quaternary nitrogen appeared and the intensity of N-H peaks sharply decreased, meaning that tertiary amines were converted to quaternary ammonium. The bands at 2960 cm-1 are assigned to C-H antisymmetric and symmetric stretching of –CH2-, indicating successful introduction of alkyl chain from 1-bromohexane.

3# 3.2.3. TG Analysis: The utility of these experimental results should be highlighted more.

Answer: Thank you for your kind suggestion, we have rewritten the discussion of TG Analysis.

The weight loss below 100 °C was due to the loss of water in the samples. The weight loss of the origin cotton linter happened in the range from 340 °C to 420 °C, which showed a one-step weight loss. The weight loss of CLF-g-GMA occurred at 220 °C and terminated at 420 °C, which was due to the complex thermal decomposition of the cotton linter and grafted epoxy groups and ester in GMA. The decrease of the degradation temperature for cellulose after grafting of GMA also happened in reference. But for CLF-g-GMA-PEHA and QCLF samples, the thermal degradation curves have two districts. The main decomposition temperature occurred between 220 and 450 °C, the second loss weight took place at 450 °C, which might be due to the residual mass of solid carbon. It suggested that the adsorbents have good thermal resistance for application in phosphate removal.

4# Line 157: “The origin phosphate concentration at pH 7…” How was obtained this value of pH???

Answer: Thanks for your kind suggestion, we have revised it. 

In this paper, pH was adjusted with 0.1 mol/L HCl or 0.1 mol/L NaOH. Because the origin pH of NaH2PO4 solution without pH adjustment was about 7, which was in the range of 4-8, so the phosphate concentration without pH adjustment was used for further batch and column adsorption tests. 

5# 3.3.3. Adsorption kinetics: The experimental results included in this section should be more detailed interpreted. 3.3.4. Adsorption isotherms: The same observation as above. 3.4. Column experiments: Again, the experimental results included in this section are too brief presented. Some additional comments/observation should be included to obtain a properly interpretation of these results.

Answer: Thanks for your kind suggestion. We have interpreted the experimental results more detail. Reference [39] have been added.

Temkin model have also used to fit the experimental data of the adsorption isotherms. We tried our best to improve the manuscript and made some changes in the manuscript follow the comments and the similarity report. These changes will not influence the content and framework of the paper. And here we did not list the changes but marked in red in revised paper.

Once again, Thanks very much for your comments and suggestions. We are looking forward to your positive response.

Thanks and best regards.

Sincerely yours,

Long Zhao

Round 2

Reviewer 2 Report

All my previous remarks and comments have been considered in this new version of the manuscript. From my point of view, the revised manuscript meets the criteria and can be published as original paper in Materials.